# The Application of a Two-Stage Decision Model to Analyze Tourist Behavior in Accommodation

**Tzong-Shyuan Chen and Chaang-Iuan Ho \***

Department of Leisure Services Management, Chaoyang University of Technology, Taichung 413310, Taiwan; jsmike@cyut.edu.tw
\* Correspondence: ciho@cyut.edu.tw

**Abstract:** As tourism products are not necessities for people's livelihood, zero consumption data are usually observed while conducting studies on topics that are relevant to tourism expenditure using cross-sectional research data, and a similar problem exists in tourist accommodation expenditure. This study adopts a two-stage process to examine the factors influencing tourist accommodation decisions in the domestic market, applying the dependent double-hurdle (DDH) model while using the dataset on Survey of Travel by R.O.C. (Taiwan) Citizens for the years 2014–2018. The findings reveal that, in the two decision-making equations, the social stratum, family life cycle, residential area, tourism behavior, vacation policy, and economic variables have different degrees and directions of influence on the intention to use and expenditure on tourist accommodation. Such information presents the processes involved in deciding to accommodate and how much to spend on accommodation, thereby indicating that it is inappropriate to use the single-equation analysis consisting of zero consumption expenditure data and to assume that the same variables influence the participation and consumption decisions.

**Keywords:** two-stage decision model; zero expenditure; dependent double-hurdle model; demand for accommodation

## 1. Introduction

Tourism is a major force in global trade that plays a vital role in the social, cultural, and economic development of most nations (Smith 1995). According to statistics compiled by the World Travel and Tourism Council, in 2019, the scale of the global tourism industry reached USD 8.9 trillion, with a contribution rate of 10.3% to the world's gross domestic product. At the same time, the industry employed 330 million people worldwide, accounting for approximately 10% of global employment. A country's tourism market generally consists of two markets with different customer sources, namely, inbound and domestic tourism. The domestic tourism market gradually expands with economic growth, increases in residents' income, and adjustments to vacation arrangements. According to the World Tourism Organization, the scale of the domestic tourism market is 10 times that of the international market (Page et al. 2001). Therefore, domestic tourism contributes significantly to a country's tourism revenue.

If one considers the example of Taiwan, in 2019 the number of inbound tourists reached 11.86 million, of which 90% were from within Asia, and the tourism revenue amounted to USD 14.411 billion (Tourism Bureau, Ministry of Transportation and Communications 2020). There were 169 million domestic travelers, 14.24 times the number of inbound tourists, although the tourism revenue was USD 12.698 billion, or 88 percent of that for the inbound tourism market. The key reason for the substantial disparity in the number of tourists despite identical revenue levels was the differences in tourist behavior between the two tourism markets. The average length of stay of inbound tourists was 6.20 nights, whereas that of domestic tourism was mainly 1.51 days, with 66% choosing to return the same day without staying in accommodation facilities. The low level of demand for accommodation

was the main reason why the performance of domestic tourism failed to surpass that of inbound tourism. Therefore, understanding the factors influencing the demand for accommodation on the part of domestic tourists in order to increase the duration of stay is an important topic when it comes to expanding the domestic tourism market.

When establishing an econometric model to discuss the factors influencing the demand of domestic tourists for accommodation, the first issue is to deal with a large influx of tourists who do not spend any money on accommodation. The traditional least squares method assumes that dependent variables have continuity and can be measured. If this approach is used to estimate model parameters when observed values are constrained by censored data, it may result in such parameters being biased and inconsistent (Maddala 1983; Judge et al. 1988). As tourism is not necessary for livelihood, the phenomenon of zero expenditure widely exists in research on tourism spending (Dardis et al. 1994; Hong et al. 1996; Cai 1999; Lee 2001; Zheng and Zhang 2013; Weagley and Huh 2004; Nicolau and Màs 2005; Jang and Ham 2009; Alegre et al. 2013; Bernini and Cracolici 2015; Sun et al. 2015). This fact makes the choice of appropriate econometric techniques crucial for the consistency of the empirical results (Maddala 1983; Amemiya 1984). With regards to zero expenditure in tourism, the models commonly used by scholars include the double-hurdle (DH) model (Cragg 1971) and the Heckit model (Heckman 1979). Unlike traditional economic models that consider the purchase and consumption decisions of consumers to occur simultaneously, these two models divide consumer behavior into two decision-making processes, i.e., whether to buy and how much to buy—also referred to as the two-stage decision model. According to the two-stage decision model that is in line with the theory of consumer behavior, consumers will collect information before purchasing products and will use that information as a reference to decide whether or not to buy, and then decide how much to spend once they have made their purchase decision.

Past studies on tourism expenditure reveal that a few of the discussions focus on the demand for tourist accommodation, for example, Hong et al. (1996) and Cai (1999). However, while both studies have adopted the Tobit model that considers zero expenditure as no consumption (Su and Yen 1996), they neglect the fact that no consumption may be the result of a lack of willingness to participate. Thus, using the Tobit model to analyze tourist accommodation expenditure may have certain limitations, resulting in an inability to grasp different influencing factors between the intention to make use of and the decision to actually spend money on tourist accommodation. More recently, a few studies have discussed this issue by using a different approach. For example, Masiero et al. (2015) utilized a quantile regression model to analyze the relationship between key travel characteristics and the price paid to book the accommodation. Ismail et al. (2021) adopt a two-step Chi-square automatic interaction detection (CHAID) procedure to segment spending on accommodation for visitors according to demographic, trip-related, and psychographic factors.

Accommodation is a major component of tourist expenditure (Laesser and Crouch 2006). However, in the case of domestic tourism, accommodation may not be made use of by everyone, i.e., not all individuals participate in this expenditure activity, thus reporting values of expenditure equal to zero. Therefore, the analytical tool should be adequate to account for a large proportion of observations with a value of accommodation expenditure equal to zero. This study considers a data-oriented approach, employs the nonnested test method and selects an appropriate two-stage decision model to discuss the factors influencing the consumer behavior of domestic tourists in regard to accommodation. By estimating the double-hurdle model, the effects of the associated determinants on the intention to use tourist accommodation and expenditure decisions can be identified. Furthermore, despite numerous empirical studies that examine the determinant factors of total tourism expenses, a particular determinant factor may have varying impacts on a specific expenditure type. The research results may help to improve the economic benefits of the domestic tourism market and serve as valuable reference for relevant businesses in developing marketing strategies.

## 2. Literature Review

### 2.1. Studies on Tourism Expenditure Using the Tobit Model

In past empirical studies, the Tobit model was the first model to be applied (Tobin 1958) to discuss the phenomenon of zero expenditure. Hong et al. (1996) used consumer expenditure survey data for the United States in 1990 and adopted the Tobit model to discuss the factors influencing accommodation expenditure in relation to family trips. Cai (1999) used consumer expenditure survey data for the United States in 1993 and investigated 3176 households while adopting the Tobit model to discuss the relationship between family characteristics and accommodation expenditure in leisure tourism. In the Tobit model, zero expenditure represents a true corner solution, whereas other possible factors causing zero expenditure are ignored. Other studies on tourism expenditure using the Tobit model include those by Dardis et al. (1994), Lee (2001), and Zheng and Zhang (2013).

### 2.2. Studies on Tourism Expenditure Using the DH Model and the Heckit Model

A few researchers have also employed the DH model or the Heckit model in studies on tourism expenditure. Weagley and Huh (2004) used the DH model to discuss the factors influencing the leisure expenditures of retired and near-retired households in the United States. Nicolau and Màs (2005) decomposed the tourist choice process into two stages using the Heckit model, namely, taking a holiday and holiday expenditure. They found that the expenditure decision is correlated with that of taking a holiday. Jang and Ham (2009) used the Consumer Expenditure Survey (CES) and performed Heckman's DH analysis to provide information on the two-step process for making travel consumption decisions. Alegre et al. (2013) examined Spanish household tourism participation and expenditure decisions by adopting a Heckit model. By means of the hurdle model, Bernini and Cracolici (2015) analyzed two stages of the tourist decision process: whether or not to participate in the domestic and overseas tourism markets in Italy and how much to spend. The DH model has also been applied in relation to expenditure on dining out (Jang et al. 2007).

### 2.3. Studies on Tourism Expenditure Using Other Models

In recent years, in order to better understand tourists' expenditure behavior, some researchers have employed new modeling frameworks to perform in-depth analyses. D'Urso et al. (2020) propose the fuzzy double-hurdle model, which combines the double-hurdle model with fuzzy set theory to take into account the effect of satisfaction on tourists' expenditure behavior. The new model allows the researchers to handle the imprecision of both collected information (i.e., levels of satisfaction) and the kind of measurement used (i.e., a Likert-type scale). Pellegrini et al. (2021) investigated tourists' expenditure behavior by implementing a framework that jointly adopts the stochastic frontier (SF) regression and multiple discrete–continuous extreme value (MDCEV) models. This framework allows the researchers to not only identify the maximum level of spending that the individual is willing to incur but also to assess two interrelated decisions: whether to allocate a budget for a specific expenditure category as well as the amount to be spent on that chosen category. Besides, a conditional quantile regression model has been applied in identifying leisure tourism expenditure patterns (e.g., Alfarhan et al. 2022).

In addition, other explanatory factors that may influence tourists' decision-making have been considered using various analytical techniques. Park et al. (2020) applies different estimation procedures, namely, ordinary least squares (OLS), two-stage least squares (2SLS), the Heckit model, and quantile regression (QR) to perform an analysis of the determinant factors in relation to total expenses. The role of information sources in predicting travel spending behaviors represents new possibilities for analyzing the determinants of expenditure by using QR. Chulaphan and Barahona (2021) investigated the determinants of tourist expenditure per capita in Thailand by utilizing an autoregressive distributed lag model (ARDL) and using panel-estimated generalized least square (EGLS). Such knowledge is essential for tourist authorities to develop profitable and sustainable

tourism projects in destinations whose natural resources have been affected by profit-seeking tourism.

### *2.4. Proposed Research Framework*

According to the two-stage decision model, the decision on the intention to use tourist accommodation and that of accommodation expenditure constitute the consumer behavior of tourist accommodation. Based on a summary of the previous literature on tourism expenditure (e.g., Dardis et al. 1981, 1994; Cai 1999; Nicolau and Màs 2005; Sun et al. 2015) and by considering the implementation of vacation policy, the variables influencing the intention to use and actual expenditure on tourist accommodation can be classified into six categories, namely, the economic factor, social stratum, geographical location, family life cycle, tourism behavior, and vacation policy. In this study, it is assumed that the economic factor influences the expenditure on tourist accommodation but does not influence the intention to use accommodation. This is mainly because if the same explanatory variable is included in the two sets of decision equations, it may be impossible to correctly identify the model's parameters (Newman et al. 2001). Therefore, it is necessary to add certain exclusion restrictions (Jones 1992; Newman et al. 2001; Aristei et al. 2008) to facilitate the estimation of the parameters in the model equations. In terms of the empirical application, it is usually assumed that the participation equation is a function of noneconomic factors; thus, the economic factor can be excluded from this equation (Newman et al. 2001; Aristei et al. 2008). The research framework of this study is presented in Figure 1. The research hypotheses are presented as follows.

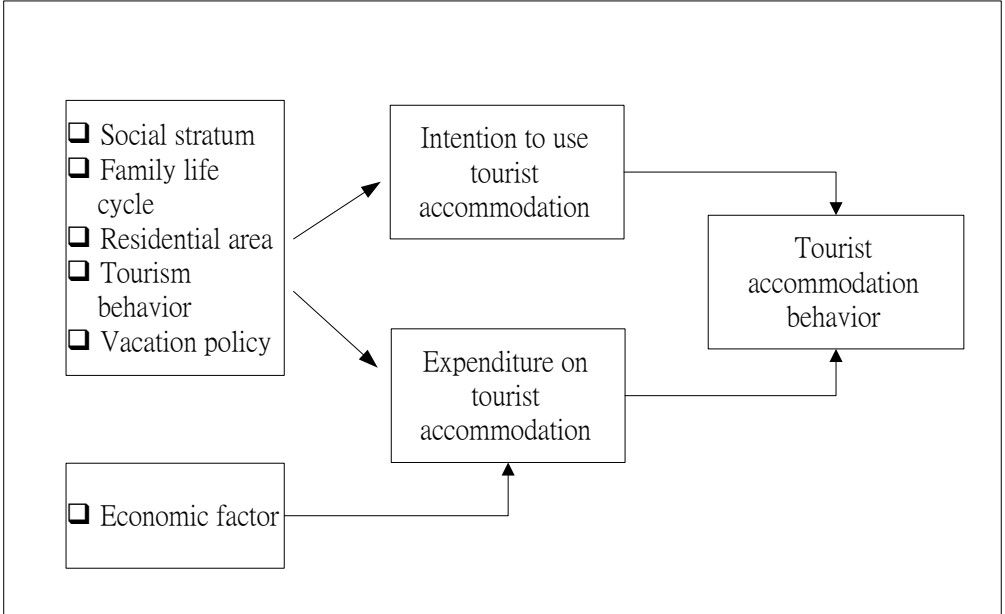

**Figure 1.** The research framework for the two-stage decision model of the intention to use and consumption expenditure on tourist accommodation.

### 2.4.1. Participation Decision

According to Nicolau and Màs (2005), Jang and Ham (2009), Alegre et al. (2013), and Bernini and Cracolici (2015), there is a positive link between the tourism participation decision and an individual's education level. Indeed, higher educational levels may provide training and preparation for some types of recreational activities (Dardis et al. 1981) and also easier access to information and knowledge (Cai 1998). Such information and knowledge are likely to increase the desire to discover new destinations and enjoy new experiences (Bernini and Cracolici 2015). Furthermore, individuals with a high level of education are more likely to reach adequate job positions and a higher level of income, which could be spent on non-basic needs like tourism. Occupation status was found to be a significant



social discriminating factor in tourism participation (Bernini and Cracolici 2015). Thus, we propose the following hypothesis:

**Hypothesis H1a.** *The social stratum has a significant impact on the intention to use tourist accommodation.*

In Jang and Ham's (2009) study, the variables of age and marital status were found to be significant for the travel decisions of elderly seniors. The research findings of Alegre et al. (2013) indicated that a positive effect was detected for tourism participation in the case of the presence of children in the household. Bernini and Cracolici (2015) found that the tourism participation decision was affected by cohort effects: the oldest cohorts were more inclined to participate in tourism than the youngest ones. The empirical results of Sun et al. (2015) indicated that the family travel intention varies at different stages of the household life cycle. Therefore, we hypothesize the following:

**Hypothesis H1b.** *The family life cycle has a significant impact on the intention to use tourist accommodation.*

By referring to Cai (1998), Nicolau and Màs (2005), Jang and Ham's (2009), and Bernini and Cracolici (2015), the empirical analysis has emphasized the role of population location and consequently the attributes of the tourists' region of residence. These studies have found that geographical variables are significant to the tourism participation decision. In a wider sense, the residential area takes in both territorial differences in tourism resources and socio-economic differences among residents' living conditions. Therefore:

**Hypothesis H1c.** *The residential area has a significant impact on the intention to use tourist accommodation.*

Four variables have been selected to represent tourism behavior, including days of the trip, travel season, travel date, and favorite activity during the trip. Li et al. (2021) revealed that tourists' behaviors in selecting travel seasons and the associated trip duration were influenced by a few factors and the correlation between these two tourism decisions was conditional upon the covariates. Dellaert et al. (1998) argued that tourists may be restricted by school holidays when choosing the period in which to travel. Indeed, time factors, including time convenience, were the most often cited reasons for not participating in recreational tourism (McGuire 1984). The finding of Wu et al. (2011) indicated that time constraints reduced the number of long trips, the number of short trips, and, to a greater extent, travel intention.

Tourists expect to recover more completely during a vacation by removing themselves from daily settings and actively engaging in various restful activities. Laybourn (2004) stated that the decision-making of festival participants may be associated with personal factors, such as lifestyle. Nicolau and Màs (2005) concluded that a greater propensity to go on holiday was associated with a favorable opinion of going on holiday. Both lifestyle and tourists' favorable opinions may reflect on their engagement in a certain activity which implies the benefits they seek (Moscardo et al. 1996). Those who seek more benefits from leisure and recreational activities may tend to lay emphasis on the high quality of travel and the use of accommodation. Thus, we hypothesize the following:

**Hypothesis H1d.** *Tourism behavior has a significant impact on the intention to use tourist accommodation.*

A vacation has been regarded as a basic human right which involves time off from work by the United Nations since 1948 and by the World Tourism Organization since 1980. In China, a vacation has been recognized as a form of human welfare (Chen et al. 2013). Vacation policies reflect the economic prosperity of a nation and have been classified into three categories: regulations regarding public holidays, regulations regarding weekly working hours, and regulations regarding paid holidays (Richards 1999). According to Chen et al. (2013), Chinese people legally have over 115 days off from work each year,

including 104 days of weekends and 11 days of vacation; in addition, employees enjoy 5 to 15 days of paid annual leave.

In Taiwan, the vacation policy changed from the original "labor has one fixed day off weekly" to "one fixed day off and one flexible rest day" in 2017. A fixed day off is compulsory to cap the number of consecutive workdays for the protection of employees' physical and mental health. A cycle of 7 days shall contain at least 1 fixed day off, and employees are not allowed to work more than 6 consecutive days unless otherwise specified. The finding of Zhang et al. (2016) indicates that the vacation policy changes adopted in China in 2007 have had a significant effect in changing the domestic tourism demand. When Taiwan has adopted a new vacation policy, it is possible that there may be a causal link between the demand for domestic tourism and the vacation policy attributes. Thus:

**Hypothesis H1e.** *The vacation policy has a significant impact on the intention to use tourist accommodation.*

2.4.2. Consumption Decision

Apart from the effects that the variables have on the decision to participate in tourism, Nicolau and Màs (2005), Jang and Ham (2009), and Alegre et al. (2013) have found evidence of a positive relationship between higher educational levels and greater tourism expenditures. As a matter of fact, those with higher levels of education are more likely to have the chance to obtain a good job and to provide their family with the opportunities to expend more money on tourism. Thus, we propose the following hypothesis:

**Hypothesis H2a.** *The social stratum has a significant impact on the expenditure on tourist accommodation.*

In the study by Nicolau and Màs (2005), the variables of age and marital status were found to have an effect on the level of tourism expenditure. The empirical study by Alegre et al. (2013) showed that the presence of children in the household had a positive effect on tourism demand, thereby increasing the household's tourism expenditure. Therefore, we hypothesize the following:

**Hypothesis H2b.** *The family life cycle has a significant impact on the expenditure on tourist accommodation.*

The geographical location of the household is an important factor influencing tourism expenditure (Dardis et al. 1981; Hong et al. 1996; Cai 1999; Zheng and Zhang 2013). Nicolau and Màs (2005) found that a long distance between the origin and destination leads to long holidays and, in turn, to higher expenditure. It also tends to result in money being spent on accommodation. Thus, we propose the following hypothesis:

**Hypothesis H2c.** *The residential area has a significant impact on the expenditure on tourist accommodation.*

Travel characteristics play a significant role in determining expenditure, such as the number of nights away (Jang et al. 2004). Among the variables related to tourism behavior, the days of the trip and the travel date are associated with trip duration, which is based on the condition of the initial decision of whether to take a trip or not. The engagement of activities may be related to how much time is spent on recreation and location (Lee 2001). These also influence itinerary planning and accommodation arrangements. Thus, we hypothesize the following:

**Hypothesis H2d.** *Tourism behavior has a significant impact on the expenditure on tourist accommodation.*

With regard to the vacation policy/tourism expenditure relationship, it is logical to assume that, once the initial decision to travel has been taken; individuals or families spend more on tourism expenditure, given that the related services required are greater

(Zhang et al. 2016). Likewise, the need for accommodation and related services may contribute to more expenditure. Thus,

**Hypothesis H2e.** *The vacation policy has a significant impact on the expenditure on tourist accommodation.*

In line with past studies (e.g., Nicolau and Màs 2005; Alegre et al. 2013), income influences tourism consumption patterns. A positive relationship between income and tourism expenditure has been identified. Thus, the hypothesis is as follows:

**Hypothesis H2f.** *The economic factor has a significant impact on the expenditure on tourist accommodation.*

### 3. Methodology

#### 3.1. Two-Stage Decision Model

The two-stage decision model is comprised of limited dependent variable models of the participation decision and consumption decision, primarily the DH model (Cragg 1971) and the Heckit model (Heckman 1979). Cragg (1971) recognized that zero expenditure may be caused by consumers choosing not to participate in the decision-making stage or choosing to participate in the first stage, but not actually spending due to certain factors when it comes to the consumption decision. In other words, the observed values for zero expenditure in the DH model not only exist in the participation decision stage but also in the consumption decision stage. According to Heckman (1979), zero spending occurs predominantly during the participation stage, with positive consumption expenditure occurring once consumers make a purchase decision.

#### 3.1.1. DH Model

The idea behind the DH model is that a consumer has to overcome two hurdles before recording a positive expenditure. These two hurdles are: (1) the participation market (potential consumers), and (2) actual consumption (Angulo et al. 2001). A complete DH model consists of the participation and consumption decisions, with equations set as follows (Jones 1989; Aristei et al. 2008):

Observed consumption:

$$Y_i = D_i * Y_i^{**} \tag{1}$$

Participation decision:

$$D_i^* = Z_i \alpha + \mu_i \, , \, \mu_i \sim N(0,1) D_i = 1 \; if \; D_i^* > 0$$
$$D_i = 0 \, , \; else \tag{2}$$

In Equation (2), a value of $D_i^*$ larger than 0 and a value of $D_i$ of 1 indicates that consumers decide to participate in the consumption. A value of $D_i^*$ equal to or less than 0 and a value of $D_i$ of 0 indicates that consumers will decide not to participate in the consumption. $Z_i$ is a variable influencing the participation decision.

Consumption decision:

$$Y_i^* = X_i \beta + v_i \, , \, v_i \sim N\left(0, \sigma^2\right)$$
$$Y_i^{**} = Y_i^* \; if \; Y_i^* > 0$$
$$Y_i^{**} = 0, \; else \tag{3}$$

In Equation (3), $Y_i^*$ is the latent consumption variable and $X_i$ is the variable influencing consumption expenditure. It can be clearly observed from Equations (2) and (3) that zero expenditure can appear in the participation decision stage when consumers choose not to participate or else choose to participate but do not have actual consumption expenditure.

Assuming that the error terms of the participation decision and consumption decision equations are mutually independent, the log-likelihood function of the independent DH model can be expressed as follows (Moffatt 2005; Aristei et al. 2008):

$$lnL = \sum_0 ln\left[1 - \Phi(Z_i\alpha)\Phi\left(\frac{X_i\beta}{\sigma}\right)\right] + \sum_+ ln\left[\Phi(Z_i\alpha)\frac{1}{\sigma}\phi\left(\frac{Y_i - X_i\beta}{\sigma}\right)\right] \tag{4}$$

In Equation (4), $\Phi(.)$ is the cumulative distribution function, $\phi(.)$ is the standard normal density function, 0 means zero consumption, and + means that the consumption value is positive.

Assuming that the error terms of the participation and consumption decision equations are correlated and that simultaneous participation and consumption decisions are possible, the bivariate normal distribution of the error terms of the two equations of the DDH model is as follows:

$$\begin{pmatrix} \mu_i \\ v_i \end{pmatrix} \sim \left[\begin{pmatrix} 0 \\ 0 \end{pmatrix}, \begin{pmatrix} 1 & \rho\sigma \\ \rho\sigma & \sigma^2 \end{pmatrix}\right] \tag{5}$$

In Equation (5), $\rho$ is the degree of correlation between the error terms of the participation and consumption decision equations. After adding the correlation coefficient, the log-likelihood function of the DDH model is as follows (Jones 1992):

$$lnL = \sum_0 ln\left[1 - \Phi\left(Z_i\alpha, \frac{X_i\beta}{\sigma}, \rho\right)\right] + \sum_+ ln\left[\Phi\left(\frac{Z_i\alpha + \frac{\rho}{\sigma}(Y_i - X_i\beta)}{\sqrt{1 - \rho^2}}\right)\frac{1}{\sigma}\phi\left(\frac{Y_i - X_i\beta}{\sigma}\right)\right] \tag{6}$$

The data distribution of limited dependent variables often reveals a significant positive skew, which is therefore unable to fulfill the hypothesis of a normal distribution of error terms. Therefore, if the maximum likelihood method is used to estimate the model, it is not possible to maintain parameter consistency. Through the inverse hyperbolic sine (IHS), dependent variables can generate consistent parameter estimates for model estimation (Newman et al. 2001). The IHS conversion function is as follows:

$$T(\theta Y_i) = log\left[\theta Y_i + \left(\theta^2 Y_i^2\right)^{1/2}\right]/\theta = sinh^{-1}(\theta Y_i)/\theta \tag{7}$$

After the dependent variables are converted through the IHS, the log-likelihood function of the DH model can be expressed as follows:

$$lnL = \sum_0 ln\left[1 - \Phi(Z_i\alpha)\Phi\left(\frac{X_i\beta}{\sigma}\right)\right] +$$
$$\sum_+ ln\left[\left(1 + \theta^2 Y_i^2\right)^{-\frac{1}{2}}\Phi(Z_i\alpha)\frac{1}{\sigma}\phi\left(\frac{[T(\theta_i Y_i) - X_i\beta]}{\sigma}\right)\right] \tag{8}$$

When using the DH model, different explanatory variables can be chosen for the participation and consumption decision equations (Jones and Yen 2000; Martínez-Espineira 2006). Early studies that applied the DH model were on cigarette and tobacco expenditures (Jones 1989, 1992; Garcia and Labeaga 1996; Aristei and Pieroni 2008) and alcoholic beverage expenditures (Angulo et al. 2001). Over the past few years, the model has been applied in a variety of fields, such as expenditure on cumulative loans (Moffatt 2005), meat products (Jones and Yen 2000; Newman et al. 2001), and nonmarket financial evaluation (Clinch and Murphy 2001; Martínez-Espineira 2006; Okoffo et al. 2016).

### 3.1.2. Heckit Model

Heckman (1979) proposed a two-step estimation method to resolve the problem of sample selection bias caused by using observable sample data. The two-step estimation method first uses the probit method to estimate the coefficients of all observed values and calculates the inverse Mills ratio (IMR). It has subsequently used the ordinary least squares method to estimate nonzero observed values, to include the IMR as an explanatory

variable, and to estimate the coefficients of the model. The Heckit model mainly comprises a selection equation and an outcome equation:

Selection equation:

$$d_i^* = z_i\alpha + \mu_i \, , \ u_i \sim N(0,1) \tag{9}$$

$$\begin{aligned} d_i &= 1 \ if \ d_i^* > 0 \\ d_i &= 0, \ else \end{aligned} \tag{10}$$

In Equation (9), $d_i^*$ is the latent variable, $z_i$ is the explanatory variable influencing participation and consumption, and $\alpha$ is the corresponding coefficient. Equation (9) reflects the relationship between $d_i^*$, the latent variable of the selection mechanism, and $d_i$, the dichotomous dummy variable actually observed (Huang and Wang 2016).

Outcome equation:

$$y_i^* = x_i\beta + v_i \, , \ v_i \sim N\left(0, \sigma^2\right) \tag{11}$$

$$y_i = y_i^* \ if \ d_i = 1 \tag{12}$$

In Equation (11), $y_i^*$ is the latent consumption expenditure variable, $y_i$ is the observed consumption expenditure variable, $x_i$ is the variable influencing consumption expenditure, and $\beta$ is the corresponding coefficient. The Heckit model assumes that the error terms ($\mu_i$ and $v_i$) of the selection equation and the outcome equation are correlated, with the degree of correlation being expressed by $\rho$. The normal distribution of the error terms of the two equations is represented in Equation (5).

Apart from the two-step estimation method, the Heckit model can also adopt the maximum likelihood method to estimate the parameters, and its log-likelihood function is as follows (Aristei et al. 2008; Wodjao 2007):

$$lnL = \sum_0 ln[1 - \Phi(z_i\alpha)] + \sum_+ ln\left[\Phi\left(\frac{z_i\alpha + \frac{\rho}{\sigma}(y_i - x_i\beta)}{\sqrt{1-\rho^2}}\right)\frac{1}{\sigma}\phi\left(\frac{y_i - x_i\beta}{\sigma}\right)\right] \tag{13}$$

### 3.2. Description of Data and Variables

This study employs domestic tourism data from the "Survey of Travel by R.O.C Citizens" conducted by the Tourism Bureau of the Ministry of Transportation and Communications of Taiwan from 2014 to 2018. The sample covers 60,817 individuals, with 26,085 having tourist accommodation and an average accommodation expenditure of NTD 1824. As for the dependent variables, the discrete nature of the decision "having accommodation" is represented as a dichotomous variable, in such a way that it takes a value of 1 if tourists have accommodation, and 0 if otherwise. This variable, related to accommodation expenditure, is found by a quantitative variable that represents the cost incurred during the accommodation. The six categories of explanatory variables are described as follows.

1. Economic factor: The individual's average monthly income. This variable is divided into six categories: no income, under NTD 30,000, NTD 30,001–50,000, NTD 50,001–70,000, NTD 70,001–100,000, and over NTD 100,001 (Table 1). The group with less than NTD 30,000 in average monthly income accounts for the largest proportion at 39.0%, followed by NTD 30,001–50,000 at 27.62%.

2. Social stratum: Education level and occupation. The education level is divided into five categories, namely, elementary (junior) high school and below, senior high (vocational) school, college, university, and postgraduate school or above, with the level of elementary (junior) high school and below as the benchmark for comparison. Among the five categories of education level, university accounts for the largest proportion at 31.38%. Occupation is divided into five categories as follows: white-collar worker, blue-collar worker, housewife, retiree, and others, with the blue-collar worker as the benchmark. Among the five categories of occupation, blue-collar workers account for the largest proportion at 45.33%.

3. Family life cycle: Includes variables, such as gender, traveling companions between the ages of 7 and 11, traveling companions between the ages of 0 and 6, marital status, and age. In terms of gender, females make up the majority, accounting for 56.67%. Marital status is divided into three categories, namely, unmarried, married, divorced/ separated, or widowed, among which the married group accounts for the largest proportion at 71.49%. Age is divided into seven categories, with 20–29 as the benchmark, and the 40–49 age group accounts for the largest proportion at 22.0%. The average number of children is 0.2 for the groups "traveling with children between the ages of 0 and 6" and "traveling with children between the ages of 7 and 11."

4. Residential area: This study classifies the residential area of respondents into five regions, namely, northern, central, southern, eastern, and other regions. Among them, the northern region accounts for the largest proportion at 43.45%, with other regions being used as the benchmark.

5. Tourism behavior: Includes the days of the trip, travel season, travel date, and favorite activity during the trip. The average days for domestic trips are 1.72 days. There are four travel seasons, and individuals primarily travel in the first season, which accounts for 27.5%. The travel date is divided into national holidays, workdays, weekends, and Sundays; most individuals travel during weekends and Sundays, which accounts for 54.25%. Favorite activities during the trip include sightseeing, cultural experience, sports, visiting amusement parks, tasting food and snacks, visiting family and friends, and others. Among them, sightseeing accounts for the largest proportion at 40.45% and visiting amusement parks accounts for the smallest proportion at 2.04%.

6. Vacation policy: The Taiwanese government has implemented a leave policy that enforces a five-day work week with "one fixed day off and one flexible rest day" since December 2016.

**Table 1.** Explanatory variables, measurement method, and statistical values of decision models for the intention to use and expenditure on tourist accommodation.

| Variable | Description | Measurement Method | | Statistical Value |
|---|---|---|---|---|
| **Economic Factor** | | | | |
| DSP | No income | 1: Yes | 0: No | 13.40% |
| | Average monthly income under NTD 30,000 | 2: Yes | 0: No | 39.00% |
| | Average monthly income between NTD 30,001–50,000 | 3: Yes | 0: No | 27.62% |
| | Average monthly income between NTD 50,001–70,000 | 4: Yes | 0: No | 12.10% |
| | Average monthly income between NTD 70,001–100,000 | 5: Yes | 0: No | 4.41% |
| | Average monthly income over NTD 100,001 | 6: Yes | 0: No | 3.47% |
| **Social Stratum** | | | | |
| EDU1 | Education level of elementary (junior) high school and below | Omitted variable | | 15.06% |
| EDU2 | Education level of senior high (vocational) school | 1: Yes | 0: No | 28.92% |
| EDU3 | Education level of college | 1: Yes | 0: No | 16.37% |
| EDU4 | Education level of university | 1: Yes | 0: No | 31.38% |
| EDU5 | Education level postgraduate school or above | 1: Yes | 0: No | 8.27% |
| OCU1 | Occupation of white-collar worker | 1: Yes | 0: No | 14.56% |
| OCU2 | Occupation of blue-collar worker | 1: Yes | 0: No | 45.33% |
| OCU3 | Occupation of retiree | 1: Yes | 0: No | 12.02% |
| OCU4 | Occupation of housewife | 1: Yes | 0: No | 17.94% |
| OCU5 | Others | Omitted variable | | 10.14% |

**Table 1.** *Cont.*

| Variable | Description | Measurement Method | Statistical Value |
|---|---|---|---|
| **Family Life Cycle** | | | |
| SEX | Gender | 1: Male  0: female | 43.33% |
| A11 | The number of traveling companions between the ages of 7 and 11 | The number of traveling companions between the ages of 7 and 11 | 0.20 people |
| A06 | The number of traveling companions between the ages of 0 and 6 | The number of traveling companions between the ages of 0 and 6 | 0.20 people |
| MAR1 | Unmarried | 1: Yes  0: No | 23.67% |
| MAR2 | Married | 1: Yes  0: No | 71.49% |
| MAR3 | Divorced/separated or widowed | Omitted variable | 4.84% |
| AGE1 | 12–19 | 1: Yes  0: No | 5.93% |
| AGE2 | 20–29 | Omitted variable | 12.64% |
| AGE3 | 30–39 | 1: Yes  0: No | 16.48% |
| AGE4 | 40–49 | 1: Yes  0: No | 22.00% |
| AGE5 | 50–59 | 1: Yes  0: No | 21.65% |
| AGE6 | 60–69 | 1: Yes  0: No | 15.56% |
| AGE7 | Over 70 | 1: Yes  0: No | 5.74% |
| **Residential Area** | | | |
| RN | Resides in the northern region | 1: Yes  0: No | 43.45% |
| RC | Resides in the central region | 1: Yes  0: No | 22.85% |
| RS | Resides in the southern region | 1: Yes  0: No | 27.99% |
| RE | Resides in the eastern region | Omitted variable | 4.38% |
| RO | Resides in other regions | 1: Yes  0: No | 1.34% |
| **Tourism Behavior** | | | |
| TDS | Days of the trip | | 1.72 days |
| SEA1 | Travel season between January and March | 1: Yes  0: No | 27.50% |
| SEA2 | Travel season between April and June | Omitted variable | 24.29% |
| SEA3 | Travel season between July and September | 1: Yes  0: No | 24.45% |
| SEA4 | Travel season between October and December | 1: Yes  0: No | 23.76% |
| TD1 | National holidays | 1: The travel date is during national holidays  0: Others | 14.30% |
| TD2 | Weekends and Sunday | 1: The travel date is during weekends and Sunday  0: Others | 54.25% |
| TD3 | Workdays | 1: The travel date is national workdays  0: Others | 31.45% |
| ACT1 | Sightseeing | 1: Sightseeing is the favorite activity during the trip  0: Others | 40.45% |
| ACT2 | Cultural experience | 1: Cultural experience is the favorite activity during the trip  0: Others | 11.75% |
| ACT3 | Sports | 1: Sports is the favorite activity during the trip  0: Others | 3.10% |
| ACT4 | Amusement park activities | 1: Amusement park activities is the favorite activity during the trip  0: Others | 2.04% |
| ACT5 | Tasting food and snacks | 1: Tasting food and snacks is the favorite activity during the trip  0: Others | 12.87% |
| ACT6 | Others | 1: Other activities are the favorite activities during the trip  0: Others | 14.16% |
| ACT7 | Visiting family and friends | Omitted variable | 15.62% |

**Table 1.** *Cont.*

| Variable | Description | Measurement Method | Statistical Value |
|---|---|---|---|
| | **Vacation Policy** | | |
| HP | "One fixed day off and one flexible rest day" policy | 1: Between 2017 and 2018<br>0: No | 0.6 |

Source of data: Summarized by this study.

## 4. Results and Discussions

This study uses four two-stage decision models, namely, the Heckit model, DH model, DDH model, and IHS DH model. Moreover, it adopts the nonnested Vuong testing method to select models suitable for the demand for accommodation in domestic tourism. Vuong (1989) used the log-likelihood function value as the basis, applied simple conversion equations, and proposed modified likelihood ratio testing for the nonnested maximum likelihood estimation. This study uses *STATA* software to perform the maximum likelihood estimation for limited dependent variable models, namely, the Heckit model, DH model, DDH model, and IHS DH model. The final log-likelihood function values of various models are depicted in Table 2, and these figures are further tested via nonnested specification tests. In terms of the nonnested test for the Heckit model vs. the DH model, the Vuong value is 3.21 (Table 3), indicating that the Heckit model is significantly better than the DH model. In terms of the nonnested test for the Heckit model vs. the IHS DH model, the Vuong value is 24.18, indicating that the Heckit model is better than the IHS DH model. In terms of the nonnested test for the Heckit model vs. the DDH model, the Vuong value is $-102.78$, indicating that the DDH model is better than the Heckit model. It can be determined through a series of nonnested tests that the DDH model is significantly better than the Heckit model, DH model, and IHS DH model. Based on the above results of the specification tests, of the four limited dependent variable models, this study suggests that the DDH model is more appropriate for explaining the decision-making behaviors in relation to the intention to use and the expenditure on accommodation in domestic tourism.

**Table 2.** Maximum likelihood function values of various limited dependent variable models.

| Model | Log-Likelihood Function Value |
|---|---|
| Heckit | $-38,734.4$ |
| Double-Hurdle | $-38,811.3$ |
| Dependent Double-Hurdle | $-38,732.0$ |
| IHS Double-Hurdle | $-38,744.6$ |

**Table 3.** Specification tests.

| Model | Test Type | Test Value |
|---|---|---|
| Heckit vs. Double-Hurdle | Vuong | 3.21 |
| Heckit vs. IHS Double-Hurdle | Vuong | 24.18 |
| Heckit vs. Dependent Double-Hurdle | Vuong | $-102.76$ |

### 4.1. Results of Participation Decision

Table 4 depicts the estimated coefficients of the DDH model with regard to the decisions on the intention to use and the expenditure on accommodation in domestic tourism. The Wald test (Table 5) and Table 4 reveal that the variables for the social stratum, family life cycle, tourism behavior, residential area, and vacation policy have a significant impact on people's intention to use accommodation in domestic tourism, supporting hypotheses H1a, H1b, H1c, H1d, and H1e.

**Table 4.** Estimated coefficients of the DDH model of the intention to use and expenditure decision on tourist accommodation.

| Variable | Consumption Decision | | | Participation Decision | | |
|---|---|---|---|---|---|---|
| | Coefficient | SD | z | Coefficient | SD | z |
| DSP | 0.0369 ** | 0.0031 | 12.07 | | | |
| SEX | −0.0537 ** | 0.0098 | −5.50 | 0.0361 * | 0.0145 | 2.50 |
| EDU2 | 0.0199 | 0.0172 | 1.15 | 0.0878 ** | 0.0236 | 3.72 |
| EDU3 | 0.0373 | 0.0191 | 1.95 | 0.1719 ** | 0.0266 | 6.47 |
| EDU4 | 0.0617 ** | 0.0186 | 3.31 | 0.2220 ** | 0.0256 | 8.67 |
| EDU5 | 0.1039 ** | 0.0227 | 4.58 | 0.2205 ** | 0.0327 | 6.74 |
| OCU1 | −0.0098 | 0.0266 | −0.37 | 0.2620 ** | 0.0367 | 7.14 |
| OCU2 | −0.0110 | 0.0237 | −0.46 | 0.1713 ** | 0.0330 | 5.19 |
| OCU3 | 0.0721 * | 0.0286 | 2.52 | 0.0957 * | 0.0413 | 2.32 |
| OCU4 | 0.0997 ** | 0.0263 | 3.79 | 0.0535 | 0.0379 | 1.41 |
| A711 | −0.1303 ** | 0.0081 | −16.06 | 0.0957 ** | 0.0126 | 7.60 |
| A06 | −0.1060 ** | 0.0084 | −12.55 | 0.0336 * | 0.0132 | 2.55 |
| MAR1 | 0.0407 | 0.0283 | 1.44 | −0.0541 | 0.0407 | −1.33 |
| MAR2 | 0.0151 | 0.0232 | 0.65 | 0.0987 ** | 0.0329 | 3.00 |
| TDS | 0.3303 ** | 0.0090 | 36.50 | 0.9531 ** | 0.0075 | 127.66 |
| HP | −0.0481 ** | 0.0090 | −5.37 | 0.0974 ** | 0.0133 | 7.32 |
| AGE1 | 0.0416 | 0.0303 | 1.37 | 0.0221 | 0.0428 | 0.52 |
| AGE3 | 0.0705 ** | 0.0190 | 3.71 | −0.0038 | 0.0283 | −0.13 |
| AGE4 | 0.0693 ** | 0.0200 | 3.46 | −0.0021 | 0.0297 | −0.07 |
| AGE5 | 0.1662 ** | 0.0218 | 7.62 | −0.0553 | 0.0319 | −1.74 |
| AGE6 | 0.2009 ** | 0.0246 | 8.17 | −0.0483 | 0.0360 | −1.34 |
| AGE7 | 0.2515 ** | 0.0321 | 7.82 | −0.1516 ** | 0.0464 | −3.27 |
| SEA1 | 0.0271 * | 0.0128 | 2.12 | −0.0889 ** | 0.0190 | −4.68 |
| SEA3 | −0.0095 | 0.0123 | −0.77 | 0.0623 ** | 0.0186 | 3.34 |
| SEA4 | −0.0101 | 0.0128 | −0.79 | 0.0484 * | 0.0188 | 2.58 |
| RN | 0.1056 ** | 0.0202 | 5.21 | −0.0222 | 0.0315 | −0.7 |
| RW | 0.0178 | 0.0213 | 0.84 | 0.0443 | 0.0330 | 1.34 |
| RS | 0.0030 | 0.0208 | 0.14 | 0.0543 | 0.0324 | 1.67 |
| RO | 0.1739 ** | 0.0446 | 3.90 | −1.1776 ** | 0.0636 | −18.52 |
| TD1 | 0.0902 ** | 0.0149 | 6.03 | −0.3690 ** | 0.0222 | −16.6 |
| TD3 | 0.0398 ** | 0.0099 | 4.03 | 0.0741 ** | 0.0151 | 4.91 |
| ACT2 | 0.0578 ** | 0.0152 | 3.79 | −0.3049 ** | 0.0215 | −14.16 |
| ACT3 | 0.0716 ** | 0.0194 | 3.69 | 0.2876 ** | 0.0354 | 8.13 |
| ACT4 | 0.1740 ** | 0.0272 | 6.39 | 0.0416 | 0.0435 | 0.96 |
| ACT5 | 0.0902 ** | 0.0143 | 6.32 | −0.2617 ** | 0.0204 | −12.8 |
| ACT6 | 0.1717 ** | 0.0130 | 13.24 | −0.1328 ** | 0.0193 | −6.89 |
| ACT7 | 0.2994 ** | 0.0244 | 12.27 | −1.2333 ** | 0.0247 | −49.93 |
| Con | 2.0239 ** | 0.0517 | 39.18 | −1.5307 ** | 0.0647 | −23.64 |
| ρ | 0.6057 ** | 0.0061 | 99.12 | | | |

Note: ** represents the null hypothesis with a significance level of 1% and a coefficient of 0, and * represents the null hypothesis with a significance level of 5% and a coefficient of 0.

**Table 5.** Wald test for the DDH model.

| Variable | Participation Decision | Consumption Decision |
|---|---|---|
| Social stratum | $\chi^2_{(8)} = 236.27$ ** | $\chi^2_{(8)} = 101.18$ ** |
| Family life cycle | $\chi^2_{(11')} = 133.12$ ** | $\chi^2_{(11)} = 218.2$ ** |
| Residential area | $\chi^2_{(14)} = 17,162.87$ ** | $\chi^2_{(14)} = 3595.58$ ** |
| Tourism behavior | $\chi^2_{(4)} = 472.73$ ** | $\chi^2_{(4)} = 128.71$ ** |

Note: ** represents the null hypothesis with a significance level of 1% and all coefficients of 0.

As regards to the individual variables, we first observed the impact of the variables for the social stratum on the intention to use tourist accommodation. There is a positive relationship between the education level and the intention to use tourist accommodation with the coefficients of the variables for the four education levels being significantly differ-

ent from 0, of which the group with a university level education (EDU4) has the highest intention to use tourist accommodation in domestic tourism, followed by the group with a postgraduate school or above education level (EDU5). As for the occupation variables, the occupation of students and unemployed (OCU5) is used as the benchmark, and the variable coefficients for white-collar workers (OCU1), blue-collar workers (OCU2), and retirees (OCU3) are significantly different from 0. Through observing the estimated coefficients of the occupation variables, the white-collar group has the highest intention to use tourist accommodation, followed by the blue-collar group, indicating that employed workers have a relatively high demand for vacation and tourism quality beyond their busy schedules, whereas the group of students and unemployed has the lowest intention to use tourist accommodation. The results related to education level and occupation variables are consistent with previous studies (Nicolau and Màs 2005; Jang and Ham 2009; Alegre et al. 2013; Bernini and Cracolici 2015).

With respect to the family life cycle, females have a significantly higher intention to use tourist accommodation compared to males. The numbers of traveling companions between the ages of 0 and 6 (A06) and 7 and 11 (A711) have a significant positive impact on the intention to use tourist accommodation. In terms of the marital status variables, the married group (MAR2) has the highest intention to use tourist accommodation with a significant estimated coefficient; the unmarried group (MAR1) has the lowest intention to use tourist accommodation with an insignificant estimated coefficient. In terms of the age variables, the 12–19 age group (AGE1) has the highest intention to use tourist accommodation and the over 70 age group (AGE7) has the lowest intention to use tourist accommodation, with a coefficient that is significantly different from 0. As age increases, the intention to use tourist accommodation declines (Figure 2). With regard to the residential area, the eastern region (RE) is used as the benchmark, and among the four residential areas, only the variable coefficient for other regions (RO) reaches the significance level. From the perspective of the estimated coefficients, tourists residing in the southern region (RS) have the highest intention to use accommodation, and those residing in other regions have the lowest intention to use accommodation. The results provide proof for the argument of Jang and Ham (2009) and Bernini and Cracolici (2015) that the family life cycle, and in particular, having children in the household, is a determinant of the travel decision and, as a result, of the accommodation decision.

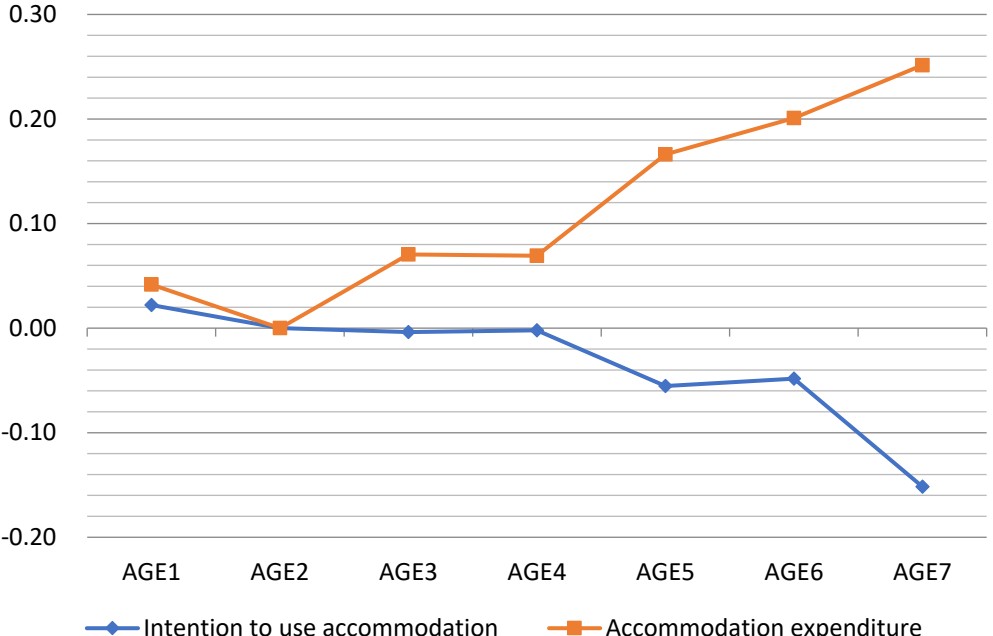

**Figure 2.** Estimated coefficients of age variables regarding the intention to use and expenditure on accommodation in domestic tourism.

In terms of the tourism behavior variables, the variable coefficients for the three travel seasons are significantly different from 0, and the third season (SEA3) witnesses the highest intention to use tourist accommodation, whereas the first season (SEA1) witnesses the lowest. Workdays (TD3) witness the highest intention to use tourist accommodation, whereas national holidays (TD1) witness the lowest intention to use tourist accommodation; the estimated coefficients of the two variables are significantly different from 0. Regarding the variables for the favorite activity during the trip, except for visiting amusement parks (ACT4), other variables are significantly different from 0; individuals who prefer sports (ACT3) and visiting amusement parks (ACT4) have a higher intention to use accommodation, whereas those who prefer visiting families and friends (ACT7) and cultural experience (ACT2) have a lower intention to use accommodation. Days of the trip (TDS) reveal a significant positive impact on the intention to use tourist accommodation. The implementation of the "one fixed day off and one flexible rest day" policy has a significant positive impact on the intention of Taiwanese to use tourist accommodation. Therefore, the vacation policy variable is a determinant of the accommodation decision, in line with Zhang et al. (2016).

Tourist accommodation, to a certain extent, reflects the importance attached by individuals to tour quality, and the single-day tour approach often sacrifices tour quality due to time constraints. The above analyses can be summarized as follows: females, people with a university level of education, white-collar workers, tourists traveling with children between the ages of 0 and 6 and 7 and 11, married people, people aged 12–19, residents of the southern region, people traveling during the third season, people traveling during normal days, and people preferring sports and visiting amusement parks are those with a high intention to use accommodation in domestic tourism.

### 4.2. Results of Consumption Decision

As for the consumption decision regarding expenditure on accommodation in domestic tourism, the economic factor, social stratum, family life cycle, residential area, tourism behavior, and vacation policy are variables with significant influence (see Tables 4 and 5). Research hypotheses H2a, H2b, H2c, H2d, H2e, and H2f are all supported. In terms of the economic factor, an individual's average monthly income has a significant positive correlation with the tourist accommodation expenditure; in other words, with an increase in income, the amount of money a family spends on tourist accommodation during domestic trips also increases. This research result is in line with the research findings by Thompson and Tinsley (1978), Dardis et al. (1981), Davies and Mangan (1992), Dardis et al. (1994), Hong et al. (1996), Fish and Waggle (1996), Cai (1999), Weagley and Huh (2004), Alegre et al. (2013), and Sun et al. (2015), i.e., there is a positive correlation between income and tourism expenditure.

In terms of the social stratum, among the education level variables, only EDU4 and EDU5 reach the significance level, indicating that there is a positive correlation between the education level and accommodation expenditure in domestic tourism. As the education level increases, the accommodation expenditure in domestic tourism also increases. Studies conducted by Dardis et al. (1981), Dardis et al. (1994), Hong et al. (1996), Cai (1999), Weagley and Huh (2004), Alegre et al. (2013), Bernini and Cracolici (2015), and Sun et al. (2015) also obtained the same result. In terms of occupation, the coefficients for retirees and housewives are significantly different from 0; housewives have the highest tourist accommodation expenditure, and blue-collar workers have the lowest tourist accommodation expenditure.

With regard to the family life cycle, the accommodation expenditure of females is higher than that of males, with a coefficient significantly different from 0. In terms of marital status, the coefficients are all insignificant; the unmarried group has the highest tourist accommodation expenditure, followed by the married group, and the divorced/separated or widowed group has the lowest expenditure. There is a significant negative correlation between the numbers of traveling companions between the ages of 0 and 6 and 7 and 11 and tourist accommodation expenditure, mainly because the higher the number of traveling companions between the ages of 0 and 6 and 7 and 11, the higher the tourism expenditure,

and thus the accommodation budget needs to be reduced. In terms of the age variables, only AGE1 is insignificant, and the other age groups are all significantly different from 0, with individuals over the age of 70 having the highest tourist accommodation expenditure. Among those over the age of 40, as age increases, the tourist accommodation expenditure also increases (Figure 2). Compared with the study by Nicolau and Màs (2005), we obtained similar results in terms of age and marital status, showing their effect on the level of accommodation/tourism expenditure. Unlike Alegre et al. (2013) who found evidence of a positive and increasing relationship with the household's tourism expenditure, we found that the accommodation expenditure behavior in Taiwan is negatively affected by the presence of children in the household.

In terms of tourism behavior, the first season witnesses the highest tourist accommodation expenditure with a coefficient significantly different from 0. The reason for this is that the first season coincides with the school winter vacation and the Lunar New Year festival, which is the peak tourism season in Taiwan, and the demand for accommodation significantly rises, thereby increasing tourist accommodation expenses. The fourth season witnesses the lowest tourist accommodation expenditure, with an insignificant coefficient. In terms of the travel date, the two variables are both significantly different from 0; national holidays witness the highest tourist accommodation expenditure, followed by workdays, and then weekends and Sundays. In terms of the favorite activity during the trip, the coefficients of all six variables reach the significance level. Individuals visiting family and friends and those visiting amusement parks have the highest tourist accommodation expenditure, whereas those engaging in cultural experience and sightseeing activities have the lowest accommodation expenditure. There is a significant positive correlation between the days of the trip and tourist accommodation expenditure, in line with the finding from Nicolau and Màs (2005), indicating that longer stays lead to higher spending levels.

With regard to residential areas, other regions witness the highest tourist accommodation expenditure, followed by the northern region, and the coefficients of both reach the significance level, with tourists residing in the eastern region having the lowest accommodation expenditure. The days of the trip (TDS) have a significant positive impact on tourist accommodation expenditure. The implementation of the "one fixed day off and one flexible rest day" policy has a significant negative impact on tourist accommodation expenditure. This might be because, following the implementation of the policy, employees of private enterprises have more vacations and more opportunities to travel overseas, thereby reducing the accommodation expenditure in domestic tourism. Zhang et al. (2016) obtained a similar finding: as China implemented a new vacation policy, the domestic tourism demand was substituted by an increasingly large outbound tourism market.

Based on the above analyses, it can be determined that females, those in high income groups, people with a postgraduate school or above education level, housewives, people traveling with fewer children between the ages of 0 and 6 and 7 and 11, people over the age of 70, people traveling during the first season, people traveling during national holidays, people who prefer visiting family members and friends and visiting amusement parks, and residents of other regions are those with higher accommodation expenditure in domestic tourism.

## 5. Conclusions and Implications

Increasing the demand for accommodation in domestic tourism is currently an important topic for developing the tourism industry, in particular when international tourism is faced with the difficulties brought about by the COVID-19 pandemic. As tourism products are not necessities for livelihood, situations where there is zero consumption and accommodation expenditure in tourism frequently occur. When conducting relevant research on tourism expenditure using cross-sectional survey data, it is necessary to incorporate zero consumption expenditure into the demand estimation model. In the discussion of tourism expenditure, it is necessary to face and deal with the issues of using appropriate analytical

models, understanding the selection process of consumption, and analyzing the factors influencing participation and consumption decisions.

This study employs a two-stage decision model to discuss the factors influencing tourist accommodation expenditure in domestic tourism in Taiwan. It considers a data-oriented approach, uses the nonnested test method and selects the DDH model as the analytical model. According to the empirical results, the participation decision to make use of accommodation in domestic tourism is influenced by five categories of variables, namely, the social stratum, family life cycle, tourism behavior, residential area, and vacation policy. The decision to engage in tourist accommodation expenditure is influenced by six categories of variables, namely, the economic factor, social stratum, family life cycle, tourism behavior, residential area, and vacation policy. The variables in the two decision equations have different degrees and directions of impact on the intention to use accommodation and to spend money on it. Therefore, it is inappropriate to use single-equation analysis consisting of zero consumption expenditure data and to assume that the same variables influence the participation and consumption decisions. This study contributes to the existing literature by being the first to attempt to apply a two-stage model specification to the accommodation decision process, that is, whether or not to use accommodation and how much to spend.

In terms of the individual variables, there is a significant positive correlation between an individual's average monthly income and tourist accommodation expenditure. There is a significant positive correlation between an individual's education level and intention to use accommodation in domestic tourism. People usually have higher-paying occupations when they have a higher education level (Nicolau and Màs 2005). With the increase in education level, the intention to use accommodation in domestic tourism increases, thereby increasing the accommodation expenditure. White-collar workers have the highest intention to use accommodation in domestic tourism, whereas students and unemployed people have the lowest intention. In terms of accommodation expenditure, housewives have the highest expenditure, followed by retirees, then students and unemployed people. Females have a higher intention to use and higher expenditure on accommodation in domestic tourism compared to males. The number of traveling companions between the ages of 0 and 6 and 7 and 11 has a significant positive impact on the intention to use accommodation in domestic tourism, but a negative impact on accommodation expenditure. While this does not mean that the number of traveling companions between the ages of 0 and 6 and 7 and 11 acts as a hindrance to accommodation in domestic tourism, in considering the limitations of their overall travel budget, those tourists may have to reduce their accommodation expenditure.

As for marital status, married people have the highest intention to use accommodation in domestic tourism, whereas unmarried people have the highest accommodation expenditure. People in the 12–19 age group have a higher intention to use accommodation in domestic tourism. As for expenditure on accommodation, for the over 40 age groups, accommodation expenditure increases with age and reaches a peak with the over 70 age group. Every year, the third season witnesses the highest intention to use accommodation in domestic tourism. With regard to accommodation expenditure, the highest amount recorded is in the first season, reflecting the seasonal features and characteristics of the domestic tourism market. In terms of the travel date, workdays witness the highest intention to use accommodation in domestic tourism, whereas national holidays witness the lowest intention to use accommodation. This could be caused by the limited accommodation supply coupled with higher expenses compared with workdays, thereby reducing the demand for accommodation. In practice, national holidays witness the highest accommodation expenditure.

In terms of favorite activities during domestic trips, the two activities of sports and visiting amusement parks have the highest intention to use accommodation in domestic tourism. By contrast, the two activities of visiting family and friends and visiting amusement parks exhibit relatively high expenditure. As for residential areas, tourists residing in the southern region of Taiwan have the highest intention to use accommodation, whereas tourists in other regions incur the highest expenditure. The "one fixed day off and one

flexible rest day" policy has a significant positive impact on the intention to use tourist accommodation, but a negative impact on accommodation expenditure.

To sum up, the results of this study indicate that accommodation expenditure models should allow for the existence of a correlation between the participation decision and the expenditure that is conditional on the participation decision. The effects of the above variables on accommodation expenditure are, however, not totally consistent with previous studies on tourism expenditure. These differences may result from the datasets, or the samples being obtained from people of different nationalities. The reasons for the differences need more investigation in future studies. Two variables, namely, tourism behavior and vacation policy, which were previously seldom included in the model's estimation, were examined in this study for their effects on the accommodation/expenditure decision. Despite the significant effects, it is necessary to more accurately understand the divergent results by performing further investigations.

Based on the analysis of the factors influencing the participation and consumption decisions in relation to domestic tourist accommodation using the two-stage decision model, the results of this research might influence the managerial direction in relation to market segmentation. Such information regarding the demand for accommodation under different economic and demographic conditions is useful to hotel managers in that it provides an alternative perspective for market segmentation. Due to the joint effect or differentiated effect of the variable, hotel managers should reconsider characterizing the profile of tourists with the greatest propensity to use accommodation and to find their expenditure patterns. This is fundamental for the development of marketing strategies. The research results lead to the following specific implications: (1) Attention could be paid to expanding the accommodation market targeted at family travelers who may consider taking children on domestic trips during the summer vacation and will choose accommodation. Therefore, entertainment and leisure space, facilities, and activities for children could be improved to develop business opportunities. (2) Faced with an aging society, there is a strong market potential for tourism for the elderly. This group has the lowest intention to use tourist accommodation but has relatively high tourist accommodation expenditure. The planning of a hospitable environment and travel itinerary for elderly travelers could be strengthened to increase accommodation incentives.

This research has some limitations. First, the model was developed and validated with data from one area. The research should be replicated to test the proposed model and hypotheses of the present research using samples from other regions and other datasets. The second limitation is that the list of variables may not be exhaustive, and thus further exploration should be encouraged. According to Isık et al. (2020), policy-related economic uncertainty plays a significant role in tourists' vacation plans. Thus, the EPU index could be included as a predictor of tourism demand. Third, the impact of the COVID-19 pandemic on travel should be a topic for further research. Tourism and travel demand were reduced to a minimum level during the period of the pandemic and domestic tourism has been the first to recover as the lockdown gradually ended. A detailed analysis of the variations in the intention to use accommodation and accommodation expenditure may be a valuable topic for future research. Finally, some researchers have broadened the knowledge of tourism expenditure by adopting a new analytical approach (e.g., Alfarhan et al. 2022; Chulaphan and Barahona 2021; Pellegrini et al. 2021). With regard to the different levels of service and nature of accommodation, many facets of accommodation expenditure decisions may need to be considered, because accommodation expenditure is not a single product but rather a number of interrelated subproducts. Tourists may additionally arrange several subset decisions within accommodation expense types, such as dining, recreational activities, and travel itineraries. In referring to Park et al. (2020), the analyses of accommodation expenditure across and within expense types could be addressed in future research. A multi-perspective view of modeling is important for gaining an enhanced understanding of tourism/accommodation expenditure patterns.

**Author Contributions:** T.-S.C. contributed to methodology, validation, formal analysis and writing-original draft. C.-I.H. contributed to funding acquisition and writing-reviewing and editing. All authors have read and agreed to the published version of the manuscript.

**Funding:** The authors acknowledge the support to this study of a grant (No. MOST 109-2410-H-324-008) from the Ministry of Science and Technology, Taiwan, R.O.C.

**Institutional Review Board Statement:** Not applicable.

**Informed Consent Statement:** Not applicable.

**Data Availability Statement:** The analyses were made based on the information contained in the datasets on Survey of Travel by R.O.C. (Taiwan) Citizens for the years 2014–2018. The information can be found in the reference list.

**Conflicts of Interest:** The authors declare no conflict of interest.

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
