# Peer review of "The Application of a Two-Stage Decision Model to Analyze Tourist Behavior in Accommodation"

_economies, doi:10.3390/economies10040071_

Round 1
Reviewer 1 Report
Thank you for the opportunity to review this manuscript. The topic of the paper is interesting, however it ham some shortcomings that I present in comments below.
Comment 1: The first two sentences in the abstract are obvious, especially the first sentence. I recommend deleting them.
Comment 2: Lines 70-71: this sentence is incorrect. There are very many articles on this subject. Apparently the author inaccurately did a literature review. Furthermore, the author only cites old references here, while there is also current literature (2020, 2021, 2022).
Comment 3: The author has indicated the state of the literature but has not done a literature review at all. Literature review section is missing. This would explain why the author claims "that only a few discussions are concerned with the demand for tourist accommodation".
Comment 4: It is completely unclear for which period the data is used.
Comment 5: I recommend extracting the discussion section from Section 3 and making this a separate section entitled "Discussion".
Comment 6: Section 4 lacks research limitations and I recommend that the added value of this paper be made more prominent.
Comment 3:
Author Response
Please see the upload file.

Reviewer 2 Report
Dear Authors,
Thank you for giving me the opportunity to read your paper. The paper “The Application of a Two-Stage Decision Model to Analyze Tourist Behavior in Accommodation” is interesting for journal readers. Kindly take note of the following specific comments to make it better.
There are several suggestions for improving your work:
Intro,
# The authors have to emphasize the main differences against the existing studies on this topic and demand site. Please visit
- https://doi.org/10.1080/13683500.2020.1734547
# The literature review should be more up to date.
# Policy implications should purely based upon empirical results.
# Need clear future recommendation/implementation in the context of economic uncertanities perspective please visit;
https://doi.org/10.1177/1354816619888346
Author Response
Please see the upload file.

Round 2
Reviewer 1 Report
Thank you for the opportunity to review this article again. The authors have taken most of the comments into account in the revised version of this paper. Unfortunately, there is still one important shortcoming.
The authors cite old references, e.g. lines 69-72. I agree that there is little or no work based on the Tobit model which considers zero expenditure as no consumption. But works which analyse the demand for tourist accommodation are plentiful and even in the last 3 years.
Reviewer 2 Report
Dear Author/s,
Thank you for giving me the opportunity to read your revised paper. I have two more minor comments to you.
# The identification of the tourism demand determinants benefits for the model are not clear.
# The literature review is still not be more up to date. Please visit following main studies about
- https://doi.org/10.1016/j.tourman.2019.01.014
I have no any other comment. Thank you
